# Parainfluenza Virus 5 Infection in Neurological Disease and Encephalitis of Cattle

**DOI:** 10.3390/ijms21020498

**Published:** 2020-01-13

**Authors:** Melanie M. Hierweger, Simea Werder, Torsten Seuberlich

**Affiliations:** 1Division of Neurological Sciences, Department of Clinical Research and Veterinary Public Health, Vetsuisse Faculty, University of Bern, 3012 Bern, Switzerland; melanie.hierweger@vetsuisse.unibe.ch (M.M.H.); simeawerder@gmail.com (S.W.); 2Graduate School for Cellular and Biomedical Sciences, University of Bern, 3012 Bern, Switzerland

**Keywords:** parainfluenza virus 5, virus discovery, encephalitis, neurological disease, cattle, genome analysis, in situ hybridization, simian virus 5, bovine astrovirus, bovine herpesvirus

## Abstract

The etiology of viral encephalitis in cattle often remains unresolved, posing a potential risk for animal and human health. In metagenomics studies of cattle with bovine non-suppurative encephalitis, parainfluenza virus 5 (PIV5) was identified in three brain samples. Interestingly, in two of these animals, bovine herpesvirus 6 and bovine astrovirus CH13 were additionally found. We investigated the role of PIV5 in bovine non-suppurative encephalitis and further characterized the three cases. With traditional sequencing methods, we completed the three PIV5 genomes, which were compared to one another. However, in comparison to already described PIV5 strains, unique features were revealed, like an 81 nucleotide longer open reading frame encoding the small hydrophobic (SH) protein. With in situ techniques, we demonstrated PIV5 antigen and RNA in one animal and found a broad cell tropism of PIV5 in the brain. Comparative quantitative analyses revealed a high viral load of PIV5 in the in situ positive animal and therefore, we propose that PIV5 was probably the cause of the disease. With this study, we clearly show that PIV5 is capable of naturally infecting different brain cell types in cattle in vivo and therefore it is a probable cause of encephalitis and neurological disease in cattle.

## 1. Introduction

A great proportion of emerging zoonotic diseases show a neurological manifestation [1,2]. Therefore, but also due to animal welfare, investigating neurological diseases in animals is an important aim for both veterinary and human medicine. Because livestock lives in close proximity to humans, is slaughtered for meat production, and plays an essential role in economy, neurological diseases in these animals are especially important to investigate.

Histopathological analysis of encephalitis cases in cattle frequently reveals an inflammatory pattern of a non-suppurative encephalitis [3,4]. The histological findings typically consist of infiltrations of lymphocytes, monocytes, and plasmacells; gliaproliferation of astrocytes and microglia; and a varying degeneration of the parenchyma. The lesions distribute in all brain areas, with different focal points and different severity. A viral etiology is strongly suggested in non-suppurative encephalitis [5]. Indeed, many viruses are known to cause encephalitis with neurological symptoms in cattle. Among others rabies virus [6,7,8], Borna disease virus [8,9], herpes virus [8,10], and Schmallenberg virus [11,12] are striking examples for neurotropic viruses. However, in a great proportion of bovine viral encephalitis cases, the etiology still remains unresolved and therefore poses a potential risk [13,14].

This gap in knowledge arose in the 1960s when neurologically diseased cattle were tested for the exclusion of rabies virus. Because of the common histological and clinical appearance and the sporadic character of all available cases, Frankhauser first summarized these cases with the term “European Sporadic Bovine Encephalomyelitis” (ESBE), in 1961 in Switzerland without detecting a causative pathogen [15]. Only few attempts to identify pathogens were made at that time [16,17]. With the onset of bovine spongiform encephalopathy (BSE) surveillance in the 1990s, up until today, this type of infection has again been identified in a frequent manner in cattle with neurological symptoms in many countries within and outside of Europe [13,14,18,19,20,21,22,23]. With a frequency of approximately 10% to 23%, ESBE is an important differential diagnosis to BSE [3,18,21]. Because of the still unknown etiology, immunohistochemical studies mostly targeting rabies virus, Borna disease virus, bovine viral diarrhea virus, chlamydia, mycoplasma, and tickborne encephalitis virus, as well as PCR studies targeting chlamydia and ovine herpesvirus 2, were performed. Unfortunately, it was not possible to provide evidence of a specific pathogen for this disease [13,14].

In the last years, our research group has focused on resolving bovine encephalitis cases of an unknown etiology. With unbiased next generation sequencing (NGS), a possibility of screening clinical samples without any previous knowledge of candidate pathogens has opened up. Using this method, we were able to detect the novel virus genotype species bovine astrovirus CH13 (BoAstV CH13) [24], which was described as bovine astrovirus NeuroS1 at the same time in American studies [25]. Later, this virus was found with different methods in about one quarter to one third of all cattle with non-suppurative encephalitis presented in Switzerland [24,26]. However, about 75% of all bovine non-suppurative encephalitis cases still remain inconclusive. In a series of further NGS studies, other viruses were also detected in bovine non-suppurative encephalitis, which strongly indicates that not just one pathogen is causative of the described pathological presentation [27,28]. However, the importance of the remaining detected viruses still has to be elucidated.

One of these detected viruses is parainfluenza virus 5 (PIV5), which was found in 3 out of 37 tested bovine brain samples but not in healthy control animals [27,28]. PIV5 was firstly discovered as a cell culture contamination of monkey kidney cells and therefore was named simian virus 5 (SV5) [29]. However, monkeys do not appear being within the natural hosts of this virus, as serological studies of wild living animals revealed [30,31]. In contrast, PIV5 was identified as a respiratory pathogen in dogs that plays an important role in the kennel cough complex [32,33,34], and therefore is often referred to as canine parainfluenza virus, especially in the veterinary context. Because the virus was isolated from many different species with unknown pathogenicity but with only very little sequence variation [35,36,37], the renaming of this virus to parainfluenza virus 5 was suggested to better emphasize the wide distribution of the virus. Even though its official name is now mammalian orthorubulavirus 5 [38], in this paper, we still use the abbreviation PIV5 to provide a better connection to the scientific context.

In 1981, PIV5 was isolated from the cerebral spinal fluid of a neurologically diseased dog and characterized ultrastructurally and serologically [39,40]. After intra cerebral inoculation in gnotobiotic puppies, these animals again developed acute encephalitis and hydrocephalus [41]. Additionally, PIV5 was found to very likely be the cause for a severe and fatal respiratory disease outbreak in calves in China [42], and was already even isolated from the brains of cattle showing neurological symptoms [43]. Together, the association of PIV5 to neurological infection in dogs and cows and the ability to induce disease in cattle prompted us to further investigate its role in bovine non-suppurative encephalitis.

Interestingly, in all three PIV5-positive brain samples, additional virus sequences were detected: The first brain sample contained bovine polyomavirus 2 (BoPyV2) and bovine herpesvirus 6 (BoHV6); in the second brain sample, BoAstV CH13 was detected; and in the third brain sample, BoPyV2, BoHV6, and BoAstV CH13 were also found [24,26,27,28,44,45]. These findings complicate the clarification of the role of PIV5 in the three cases, especially because only little is known about the interplay between different viruses in neurological infections and the uncertainty of whether only one or more viruses is crucial for disease onset, distribution, and severity.

This study mainly focused on the molecular and neuropathological characterization of PIV5 infection in bovine encephalitis of unknown origin but also investigated the apparent coinfection scenario of this virus with BoAstV CH13, BoPyV2, and BoHV6. We determined and compared the whole genome sequences of the detected PIV5s among each other to already described PIV5 sequences. For a better understanding of the contribution of each virus to disease, attempts to detect all present viruses in situ were undertaken, and comparative quantitative analyses were performed. We clearly demonstrated the replication of PIV5 in many brain cell types at sites of inflammation in one animal, and therefore, we propose that this virus is a probable cause for at least some bovine encephalitis cases of unknown origin.

## 2. Results

### 2.1. Molecular Confirmation of Viruses

Three cows showing neurological symptoms and an encephalitis of unknown origin were further investigated in this study. Information on the history and initial testing results from previous studies of these cases is provided in Table 1. In the present study, in all three tested brain samples (animal IDs 26731, 26875, and 27020), PIV5 RNA was confirmed with RT-PCR and RT-qPCR. In animals 26731 and 27020, BoHV6 was confirmed with PCR, and in animals 26875 and 27020, the RT-qPCR for BoAstV CH13 gave positive results. However, it was not possible to confirm BoPyV2 with PCR in any of the samples in the present study. Therefore, this virus was excluded for further investigations. Brain tissue samples of cattle with non-suppurative encephalitis of another origin than PIV5, BoAstV CH13, or BoHV6 served as negative controls and did not show a positive result in the tested PCRs; neither did non-template controls. In order to exclude carryover of PIV5 during brain tissue preparation between animals 26731, 26875, and 27020, animal brain tissue samples that were chronologically processed in the same way between animal 26731 and 26875 or animal 26875 and 27020 at the time of admission to our diagnostic services were tested by RT-qPCR for PIV5 but remained negative.

### 2.2. Genome Sequences

In order to determine the PIV5 sequence in each of the brain RNA samples, Sanger Sequencing and rapid amplification of cDNA ends (RACE) were performed. With these techniques, the whole PIV5 genome was sequenced from animal 26731. The genome size is 15,246 nucleotides (nt). From the brain RNA samples of animals 26875 and 27020, however, it was not possible to experimentally confirm the authentic 3′ end of the virus’ genome because the RACE failed. These genome sequences remain incomplete, with seven and three nucleotides, respectively, lacking at the 3′ end in comparison to the PIV5 sequence of sample 26731. Beyond this, the alignment of the three PIV5 sequences revealed, overall, no differences on the nucleotide level. Therefore, the sequences are summarized and referred to as PIV5 CH19-MMH. In the PIV5 CH19-MMH sequence, the usual open reading frames (ORFs) for PIV5 proteins are present, with the same order, position, and direction as in already described PIV5 strains (Figure 1 A). The order and lengths of the ORFs from 3′ to 5′ are as follows: ORF for the nucleocapsid protein (N): 1530 nt; V-protein (V): 669nt, phosphoprotein (P): 1179 nt; membrane protein (M): 1131 nt; fusion protein (F): 1656 nt; small hydrophobic protein (SH): 216 nt; hemagglutinin-neuraminidase protein (HN): 1698 nt; and large protein (L): 6768 nt. The V and P proteins are translated from the ORF, sharing the same N-terminus but differing in their C-terminus. The V protein coding mRNA is directly transcribed but that of the P protein involves polymerase stuttering and an insertion of two guanine nucleotides between nucleotide position 2338 and 2339, which results in a frameshift. The ORF of the M protein is 3 nt shorter compared to all other PIV5 genomes due to a guanine to thymine mutation at position 4269. The ORF of the F protein has the same length as other field isolates of PIV5 but is 66 nt longer than PIV5 isolates from the cell culture. The ORF for the SH protein, however, is 81 nt longer compared to the PIV5 reference strain W3A due to mutations in two stop codons at position 6435 and 6447 and therefore has an overall length of 216 nt (Figure 1B).

### 2.3. Phylogenetic Analysis

When analyzing the PIV5 CH19-MMH sequence by blastn, the closest related PIV5 strain is the strain H221, with a variation of 1.8% on the nucleotide level (Figure 1A). In a maximum likelihood tree based on selected whole genome sequences, the PIV5 CH19-MMH strain has an outstanding position but again clusters closest with canine respiratory strains and not with other bovine (PV5-BC14) or neurotropic (CPI+) PIV5 strains. These sequences are more distantly related to the PIV5 CH19-MMH strain, with an identity of 97.5% and 97%, respectively. Even though profound differences in certain ORFs can be noted, considering whole genome sequences, all described full-length PIV5 strains only show little sequence variation (Figure 2).

### 2.4. In Situ Detection of PIV5

In order to detect PIV5 in situ in formalin-fixed, paraffin-embedded (FFPE) brain material of the three animals, immunohistochemistry (IHC) and in situ hybridization (ISH) were performed. Positive slides were compared to hematoxylin and eosin (H&E) stained slides of the same region to check for correlation to signs of inflammation. The PIV5 antibody used for IHC (clone 2F11F7) is directed against the PIV5 V protein, whereas the PIV5 probe used for ISH targeted the PIV5 N protein coding region. With this, it was possible to detect PIV5 antigen and RNA in the FFPE brain tissue slides of animal 26731. The positive signal consists of granular red or brown deposits mostly in the cytoplasm of cells, which were morphologically evaluated as neurons. With this, not only the presence of PIV5 but also an active replication of this virus in neurons could be demonstrated. The neuroanatomical localization of positive labelling is in accordance with signs of inflammation, which consist of perivascular cuffs, glia proliferation, and neuronal degeneration. Inflammation is most present in the diencephalon, basal ganglia, allocortex, and brainstem. Viral antigen was found in the brainstem and the paleocortex and viral RNA in the brainstem, paleocortex, midbrain, and thalamus (Figure 3A–F). Because not all positive labelling is unambiguously associated with neurons (Figure 3G–H), double staining that combines fluorescent ISH for detecting PIV5 RNA and immunofluorescence (IF) of different cell markers for neurons, astrocytes, microglia, and oligodendrocytes was performed. With this, it was possible to elucidate the infected cell types in the brain of animal 26731 (Figure 4; negative controls: Appendix A). According to a semi-quantitative evaluation, neurons were infected most frequently with PIV5 (> 60%), whereas astrocytes were less frequently infected (30–60%). PIV5 was also detected in oligodendrocytes and microglia, but at a relatively low rate (<30%). In FFPE brain tissue slides of the other animals (26875 and 27020), however, it was not possible to detect either PIV5 antigen or RNA. Both groups of negative control animals (healthy slaughtered animal and animals with non-suppurative encephalitis of another origin) also did not show any staining for PIV5 antigen or RNA.

### 2.5. In Situ Detection of Other Viruses

Because at least one other virus was confirmed in all the brain samples, these viruses were also tested for by in situ methods. ISH with RNAscope of BoAstV CH13 gave positive results in animal 26875. Positivity is again referred to as granular brown deposits. Morphological evaluation reveals the signal to be in the cytoplasm of neurons, which is consistent with previous in situ findings. Compared to H&E stained FFPE slides, RNA also correlates with signs of non-suppurative inflammation as already described [26,45] (Figure 5). In animal 27020, however, ISH of BoAstV CH13 remained negative, even though it was found to be positive for this virus in another study [26]. In both animals that were positive for BoHV6 by qPCR (animals 26731 and 27020), the presence of BoHV6 viral nucleic acid could not be confirmed by ISH.

### 2.6. Comparison of Viral Copy Numbers

In order to compare the amount of viral nucleic acid in the three animals, the (RT)-qPCRs were standardized (Appendix A) and normalized to the (RT)-qPCRs results of reference genes. In animal 26731, a high viral load of PIV5 was detected, whereas BoHV6 was only found in small quantities. In animal 26875, a high viral load of BoAstV CH13 was detected but only small amounts of PIV5. In animal 27020, however, virus genomes were only detected in low amounts (Table 2).

## 3. Discussion

We were able to obtain full length, or nearly full length, viral genomes of PIV5 by sequencing cDNA obtained from the brain tissue of three bovine cases of encephalitis of unknown origin. The virus sequences in these three animals are identical to each other but differ from already published PIV5 sequences. Therefore, this sequence represents a new PIV5 strain and is provisory termed CH19-MMH. With a length of 15,246 nt, the PIV5 CH19-MMH genome is the same size as all representatives of the PIV5 genus and is in accordance with the rule of six. This rule states that the total number of nucleotides of virus genomes belonging to the Paramyxoviridae family is a multiple of six, which is required for successful RNA replication [46]. Overall, the viral ORFs have the same localization and length as in all other PIV5s, with the exception of those encoding the matrix (M), fusion (F), and SH proteins. The F protein does not show a constant length in different PIV5 strains. Cell culture isolates have a 66 nt shorter ORF encoding the F protein compared to most field isolates, because of a selection against hyperfusiogenic PIV5 strains in vivo [35]. The ORF of the F protein of the PIV5 CH19-MMH has the same length as field isolates, supporting an in vivo origin of this new virus strain.

In PIV5 CH19-MMH, the SH-ORF is 81 nt longer, suggesting a 27 amino acid larger SH protein, which is so far unique for PIV5 strains. The SH protein is within the most variable regions of PIV5 genomes [36] and different characteristics are described (Figure 1B). Not all PIV5 strains, for example, have an SH-ORF due to mutations both in the start and stop codons, simultaneously. The neurotropic dog strains CPI+ and CPI- as well as the other bovine strain PV5-BC14 lack an SH-ORF. The lack of the SH protein leads to a higher CPE in vitro but to an attenuated virus in vivo [47]. Together, differences in the SH protein and the phylogeny show that the CH19-MMH strain differs from known neurotropic or bovine PIV5 strains. Yet, statements regarding the origin and putative interspecies transmission events of the virus have to be seen with caution considering the overall very little sequence variations among PIV5 whole genome sequences [36,37]. To investigate the expression and the potential role of the SH protein in the disease pathogenesis of the PIV5 CH19-MMH, further studies are needed and have been initiated in our laboratory.

The finding that identical sequences were determined in the three investigated animals raises the question of whether there is a true infection in all three cattle, or whether this is a result of sample contamination. Contamination could have taken place at two time points. Firstly, when the brains of the animals were removed, contaminated tools could have been the source for a carryover of this virus. However, two animals, which were processed accordingly chronologically in between our investigated animals, clearly scored as PIV5 negative by RT-qPCR. Therefore, a contamination at this level seems very unlikely. Secondly, contamination could have taken place when processing the brains in the laboratory. This is also very unlikely as every brain is processed separately with individual equipment and the PIV5-positive results originate from different studies, which had been conducted at different time points [27,28]. The fact that nearly full-length viral genomes were sequenced also counts against a contamination of samples, as for that a massive amount of carryover had to have taken place. Because it is the first time that full PIV5 genomes were sequenced from bovine brains, it is also conceivable that viral features and therefore the viral sequence has to be very specific to lead to brain infections in cattle. It was only possible to detect PIV5 in situ in animal 26731, which proves a natural infection with this virus but not in the other two animals. The lack of positive PIV5 in situ results in animals 26875 and 27020 does not exclude natural PIV5 infection in these animals. It has to be considered that for standard procedures, only one half of each brain was formalin fixed and paraffin embedded, whereas the other half of the brains is used for moleculargenetic investigations. Infections taking place locally in the brains can therefore have a positive result in moleculargenetic methods (RT-(q)PCR, Sanger Sequencing) but a negative result in in situ methods (IHC, ISH). Furthermore, each FFPE slide only has a thickness of 3 µm, which makes it impossible to screen the whole half of the FFPE brain. This is a possible explanation for the discrepancy of the PIV5 results in animals 26875 and 27020. Inflammatory patterns can also be seen in the investigated FFPE brain slides of these two animals. This suggests either that the PIV5 infection was already almost cleared in these areas or other reasons for the inflammatory pattern, for example, infection by other viruses. Both explanations are conceivable, because the stage of infection for the investigated animals is unknown and also other viruses (BoAstV CH13 and BoHV6) were discovered in these animals [24,26,27,28,44,45]. Taken together, it will not be possible to ultimately and unambiguously clarify the questions of contamination and mismatching PIV5 results retrospectively.

PIV5 was detected in situ in different neuronal cells in animal 26731. This is not unexpected, because sialic acid is the cellular receptor for PIV5, which is ubiquitously present on cells throughout the body. PIV5 does not bind indiscriminately to sialic acid but to receptor molecules with specific bound glycoproteins and glycolipids [48,49]. However, we clearly demonstrated a broad cell tropism of PIV5 in a bovine brain without determining the specific combination of sialic acid and glycoproteins or glycolipids needed for a successful cell invasion.

Interestingly, PIV5 antigen was less abundant in situ in animal 26731 than PIV5 RNA. The discrepancy of these findings suggests the presence of the virus with only minimal replication, or at least with only minimal production of V proteins, because the antibody for viral antigen detection is directed against this protein. Alterations in the V protein lead to a persistent state of PIV5 in vitro [50] and to attenuation in vivo [51]. In a persistent state, the virus can remain in the host organism for a long time without recognition from the immune system, because no or only very few viral proteins are produced. Persistent infections with PIV5 have often been described, especially in cell cultures [52,53,54], and therefore, this is a possible explanation for the discrepancy of viral RNA and antigen in situ.

Because not only PIV5 but also other viruses were detected in every brain of the three investigated cows, we investigated these as well. BoHV6 is also known as bovine lymphotropic herpesvirus and was neither detected in high viral loads nor in situ in FFPE brain material in any sample of this study. BoHV6 has a high prevalence in the cattle population and also in heathy animals. Because it was already found in peripheral blood mononuclear cells (PBMCs) in about 74% to 91% of asymptomatic cattle [55,56], and fresh frozen brain material always contains blood, PBMCs are the most plausible source for positive BoHV6 results in qPCRs in this study. The fact that this virus has a high prevalence in healthy cattle questions the relevance of positive BoHV6 findings. BoAstV CH13, on the other hand, is found frequently in bovine non-suppurative encephalitis cases [24,26]. It was already demonstrated in situ at sites of inflammation [26,45] and therefore, is a well-known probable cause for encephalitis in cattle.

For many years, Koch’s postulates were the gold standard to reveal the causality of an infection and disease [57]. However, these postulates do not meet the needs of the time anymore as some pathogens cannot be cultivated or no suitable animal models exist. Therefore, Lipkin suggested a new hierarchy of confidence for assessing the level of causality of viral infections with disease in 2013 [58]. In this hierarchy, a possible causality is given when the pathogen is isolated from the diseased tissue, nucleic acid or proteins of the pathogen are detected, or a pathogen-specific immune response can be noted. To shift a possible causality to the next level, a probable causality, among other criteria, of the pathogen has to be demonstrated in situ with a correlation to sites of pathological lesions. A high concentration of the pathogen strengthens a probable association. We detected PIV5 in situ in animal 26731 with a clear correlation to sites of inflammation. In animal 26875, however, BoAstV CH13 was detected in situ in the same areas as inflammatory patterns can be seen. In animal 27020, no virus could be detected in situ. These results are in accordance with the quantity of viruses in the frozen brain samples. PIV5 is most readily present in animal 26731, whereas in animal 26875, BoAstV CH13 exists in the highest amount. In animal 27020, no virus was found to have a high quantity. Therefore, it can be concluded that in animal 26731, the probable cause for the non-suppurative encephalitis was PIV5, and in animal 26875 BoAstV, CH13 was probably causative for the disease. In animal 27020, no virus was identified as a probable cause of the inflammation due to a lack of in situ findings and low copy numbers of the viral sequences in the brain sample. Either this animal was killed in a later stage of disease, so that the causative virus was already mostly cleared in the brain, or other infectious reasons led to an inflammatory pattern, such as immune-modulated reactions. Whether PIV5 also plays a role in the disease outbreak in animals 26875 and 27020 due to a potential helper function, as it is known for kennel cough [59,60], or neuroinvasion results from an inflammation-mediated breakdown of the blood–brain barrier, remains to be elucidated. Determinants that allow the infection of cattle and the invasion in the brain need to be addressed in future studies. In order to investigate the route of infection and the prevalence of this virus in the Swiss cattle population, screening studies of respiratory samples, as well as further brain material of cases of bovine non-suppurative encephalitis and serological studies should be performed.

## 4. Materials and Methods

### 4.1. Samples

Three animals (animal IDs 26731, 26875, 27020) that were introduced to the Division of Neurological Sciences, Vetsuisse Faculty, University of Bern in 1998 for exclusion of bovine spongiform encephalopathy (BSE) were further investigated. The animals showed different neurological symptoms but the histopathological findings in the brains were similar. All brain tissues showed a histopathological inflammation pattern of a non-suppurative encephalitis mainly in the brainstem. Fresh frozen brain tissue samples of all three animals were already tested as being positive for parainfluenza virus 5 (PIV5) in previous studies by next generation sequencing (NGS) and/or RT-PCR [27,28]. Animals 26731 and 27020 gave additional positive results for bovine polyomavirus 2 (BoPyV2) and bovine herpesvirus 6 (BoHV6) and animals 27020 and 26875 for bovine astrovirus CH13 (BoAstV CH13) [24,26,27,28,44,45] (Table 1).

Fresh frozen brain tissue as well as native and hematoxylin and eosin (H&E) stained formalin-fixed paraffin-embedded (FFPE) brain tissue slides were available from all three animals. Fresh frozen brain tissue and FFPE brain material of cows with neurological symptoms and a non-suppurative encephalitis, but that previously tested negative for the investigated viruses (animal IDs 31292, 44285, 25018, 49331, 49669, 49768, 49305) ([27] and unpublished data), as well as FFPE brain material of a healthy slaughtered cow (animal ID 50000) served as negative controls. Additionally, fresh frozen brain tissue samples of two BSE positive cows (animal IDs: 26760, 26907), which were introduced to our division timewise in between the receipt of animals 26731, 26875, and 27020, were investigated. Fresh frozen brain tissue of an animal positive for BoHV6 and a low Cq value for that virus (animal ID 42145) [27] was used for standard curve preparation.

### 4.2. Nucleic Acid Extraction

Pooled frozen tissue samples of the brainstem, cerebrum, cerebellum, and thalamus were used for RNA extraction with TRI Reagent (Sigma Life Sciences, St. Louise, MO, USA) and for DNA extraction with the DNeasy Blood and Tissue Kit (Qiagen, Hilden, Germany), both according to the manufacturer’s instructions.

### 4.3. PCR

For RT-PCRs, extracted RNA was first reverse transcribed to cDNA with SuperScript™ III Reverse Transcriptase (Thermo Fisher Scientific Inc., Waltham, MA, USA) using random hexamers as primers. PCR assays were then performed using GoTaq^®^ Green Master Mix (Promega Corporation, Madison, USA). For the detection of PIV5 cDNA, primer pairs PIV-5_1L and PIV-5_2R (targeting the open reading frame (ORF) of the N protein) as well as PIV-5_3L and PIV-5_4R (targeting the ORF of the H protein) and PIV-5_5L and PIV-5_6R (targeting the ORF of the L protein) [27] were used. PCR for the detection of BoPyV2 DNA was done with primers BPyV3 and BPyV4 [61]. All primers and PCR conditions are listed in the Appendix A). PCR reactions were subsequently analyzed by gel electrophoresis on a 1% agarose gel. Non-template controls or animal samples tested as negative for PIV5 (49331, 49669, 49768) and BoPyV2 (26875), respectively, were carried along in each reaction.

Quantitative taq-man probe-based PCRs were run on a CFX96TM Real Time System (BioRad, Hercules, USA) using the AgPath-ID One-Step RT-PCR System (Thermo Fisher Scientific Inc., Waltham, USA) and the Path-ID qPCR Master Mix (Thermo Fisher Scientific Inc., Waltham, USA), for RT-qPCRs and qPCRs, respectively. For the detection of PIV5, a primer/probe combination targeting the N protein was designed in Geneious Software (version 9.1.8, Biomatter, Auckland, New Zealand) based on the sequence of the closest related PIV5 strain H221 (accession no.: JQ743323). A primer-probe set for detecting BoAstV CH13 was published previously ([44], CH13-A). qPCR for detecting BoHV6 was done with previously published primers and probe [56]. Primer–probe combinations as well as cycling conditions are provided in the Appendix A). Data analysis was performed in the CFX Maestro software (version 4.1.2433.1219, BioRad, Hercules, USA) with an auto calculated baseline threshold. Non-template controls or animal samples previously tested as negative for the investigated virus (PIV5: 49305; BoAstV CH13: 26731; BoHV6: 26875) served as negative controls.

### 4.4. Copy Number Determination

First, a standard curve was established for each quantitative PCR protocol. For PIV5 and BoAstV CH13, viral RNA was in vitro transcribed from plasmids containing cDNA of the full-length viral genome using the MEGAscript™ T7 High Yield Transcription Kit (Thermo Fisher Scientific Inc., Waltham, USA) according to the manufacturer’s instructions. RNA was purified by spin column chromatography using MicroSpin™ S-400 HR Columns (GE Healthcare Bio-Science, Pittsburgh, USA) as described by the manufacturer and RNA yields were determined by measurements on a QuBit Fluorometer (Life Technologies, Eugene, USA). Copy numbers were calculated considering the RNA sequences and the molecular weight per μL [62]. To establish the standard curve of the BoHV6-qPCR, first, a nested PCR was performed on DNA extracts of animal 42145 to amplify the target region for the qPCR (primers HgB551F and HgB1112R, as well as HgB580F and HgB990R [56]; GoTaq^®^ Green Master Mix (Promega Corporation, Madison, USA) according to the manufacturer’s instructions; annealing temperatures of 47 and 52 °C). The PCR product was cloned into a plasmid (TOPO TA Cloning^®^ Kit (Thermo Fisher Scientific Inc., Waltham, USA)) and amplified in One Shot^®^ TOP10 competent cells (Thermo Fisher Scientific Inc., Waltham, USA) as described by the manufacturer. Plasmid DNA concentration was measured on a QuBit Fluorometer (Life Technologies, Eugene, USA) and the copy number was calculated considering the target region’s DNA sequence and molecular weight per μL [62]. RNA and DNA standards were tested in the respective PCR protocols in 10-fold dilution series in three independent replicates and standard curves were created for all investigated viruses by blotting the copy number against the corresponding Cq values. Standard curves are provided in the Appendix A).

To normalize the amount of extracted nucleic acid of each sample, we used qPCR and RT-qPCR of reference genes: Bovine beta actin for RNA extracts and bovine 12s rDNA for DNA extracts. RT-qPCR for bovine beta actin was performed with an AgPath-ID One-Step RT-PCR System (Thermo Fisher Scientific Inc., Waltham, USA) with previously published primers and probe sequences [63]. Bovine 12s rDNA-qPCR was done with primers reported previously [64], a probe designed with Geneious Software (version 9.1.8, Biomatter, Auckland, New Zealand) based on a bovine mitochondrial sequence (accession no.: NC_006853), and with Path-ID qPCR Master Mix (Thermo Fisher Scientific Inc., Waltham, USA). (RT)-qPCR protocols, primers, and probes are listed in the Appendix A.

Every RNA and DNA sample of animals 26731, 26875, and 27020 was measured with (RT)-qPCRs of the corresponding viruses in duplicate, and the average Cq value was evaluated. The average Cq value was then normalized according to the result of the (RT)-qPCRs of the reference genes, such that the same amount of nucleic acid of each sample was taken into consideration for copy number determination. Finally, the copy number was calculated from the normalized Cq value of each sample in each (RT)-qPCR with the help of the formulas of the created standard curves.

### 4.5. Sanger Sequencing

For Sanger Sequencing of the new PIV5 strains, forward and reverse sequencing primers were designed in Clone Manager Software (version 9, Sci Ed Software) and Geneious Software (version 9.1.8, Biomatter, Auckland, New Zealand) with a distance of ~500 base pairs (bp) to each other. As a template for primer design, PIV5 NGS data obtained from Bouzalas et al. [28] were completed with the sequence of the closet related PIV5 strain H221 (accession no.: JQ743323). All sequencing primers are listed in the Appendix A). RNA extracted from fresh frozen brain material was first reverse transcribed to cDNA with SuperScript™ III Reverse Transcriptase (Thermo Fisher Scientific Inc., Waltham, USA) using random hexamers or gene specific primers as described by the manufacturer. Overlapping PCR amplicons of a maximum length of 3000 bp were generated with Q5^®^ Hot Start High-Fidelity DNA Polymerase (New England Biolabs Inc., Ipswich, USA) according to the manufacturer’s instructions. After gel electrophoresis on a 1% agarose gel, bands were cut out and purified using Wizard^®^ SV Gel and a PCR Clean-Up System Kit (Promega Corporation, Madison, USA) or NucleoSpin^®^ Gel and PCR Clean-up Kit (Macherey-Nagel, Düren, Germany) according to protocols described by the manufacturer. Sanger Sequencing was performed with 3 µL of purified PCR product in a 3730 DNA Analyzer (Thermo Fisher Scientific Inc., Waltham, USA) with the BigDye^®^ Terminator v3.1 Cycle Sequencing Kit (Thermo Fisher Scientific Inc., Waltham, USA) using the available primers within the amplified sequence and the protocol provided by the manufacturer. For PIV5s of animals 26731 and 27020, the full virus genome length was Sanger Sequenced. In the case of the PIV5 of animal 26875, only gaps and low covered areas, which remained between and within contigs after NGS, were completed or reanalyzed by Sanger Sequencing. The obtained sequencing data were analyzed in Geneious Software (version 9.1.8, Biomatter, Auckland, New Zealand).

### 4.6. Rapid Amplification of cDNA Ends

For the PIV5 3′ rapid amplification of cDNA ends (RACE), extracted RNA was first ligated to a 3′ end cordecypin-blocked and 5′ end phosphorylated adaptor sequence (DT88) [65] with the T4 RNA Ligase 1 (New England Biolabs Inc., Ipswich, USA). The reaction was carried out with approximately 1 μg RNA in 1× ligase reaction buffer and 25% PEG 800, 1 mM ATP, 20 pmol DT88, and 20 U ligase and was incubated at 37 °C for 1 h. The ligated RNA was then reverse transcribed with SuperScript™ III Reverse Transcriptase (Thermo Fisher Scientific Inc., Waltham, USA) and a primer complementary to the adaptor sequence (DT89) [65]. A semi-nested PCR was performed with adaptor- and gene-specific primers using the Q5^®^ Hot Start High-Fidelity DNA Polymerase (New England Biolabs Inc., Ipswich, USA). The 5′ RACE System Kit (Thermo Fisher Scientific Inc., Waltham, USA) was used for RACE of PIV5 5′ cDNA ends following the manufacturer’s guidelines. Nested PCR was performed with adaptor- and gene-specific primers with Taq DNA Polymerase (New England Biolabs Inc., Ipswich, USA). All sequences of gene specific primers used for 3′ or 5′ RACE are provided in the Appendix A). RACE PCR products were gel-purified and Sanger Sequenced as described above.

### 4.7. Genome Comparison

For genome analyses, the new generated PIV5 sequences were first aligned and compared to one another with Geneious Software (version 9.1.1, Biomatter, Auckland, New Zealand). Then, the new PIV5 strain CH19-MMH (accession no.: MN735204) was blasted against the NCBI nucleotide sequence database with “megablast” [66]. Additionally, the sequence was aligned and compared to all full length PIV5 genome sequences available on GeneBank on 8 November 2019 with Geneious Software (version 11.1.5, Biomatter, Auckland, New Zealand): Strain W3A (accession no.: JQ743318); 78524 (JQ743319); CPI+ (JQ743320); CPI- (JQ743321); DEN (JQ743322); H221 (JQ743323); MEL (JQ743325); MIL (JQ743326); SER (JQ743328); 08-1990 (KC237063); 1168-1 (KC237064); D277 (KC237065); KNU-11 (KC852177); PV5-BC14 (KM067467); CC-14 (KP893891); AGS (KX060176); ZJQ-221 (KX100034); CPIV-HeN0718 (KY114804); PIV5-SR (KY685075); Carina (MF170888); Rigel (MF170889); PIV5-GD18 (MG921602); CAN (MH362816); HMZ (MH370862); SH/2015/1202 (MK028670); T220 (MK423232); T263 (MK423233); T361 (MK423234); T398 (MK423235); T399 (MK423236); T434 (MK423237); M32 (MK423238); M129 (MK423239); M197 (MK423240); M293 (MK423241); N99 (MK423242); and N163 (MK423243).

### 4.8. Phylogenetic Tree

A phylogenetic tree was constructed based on the full-length nucleotide sequences of PIV5 strains PV5-BC14, KNU-11, SER, CPI+, CPI- MEL, W3A, ZJQ-221, H221, 78524, and the new sequenced PIV5 CH19-MMH in MEGA X [67]. The maximum likelihood method with 1000 bootstrap replicates was used based on the General Time Reversible model, which was chosen according to the “Find Best DNA/Protein Models” option in MEGA X [67].

### 4.9. Immunohistochemistry

All available FFPE tissue slides of animals 26731, 26875, and 27020 were stained with a monoclonal mouse anti V5 TAG antibody clone 2F11F7 (Thermo Fisher Scientific Inc., Waltham, USA). The slides were first deparaffinized and rehydrated in Xylol and 100% ethanol. Endogenous peroxidase was then blocked with 3% H_2_O_2_ in methanol for 20 min. For antigen retrieval, the slides were heated in Dako REAL Retrival Solution, pH9 (Dako Denmark A/S, Glostrup, Denmark) for 20 min at 95 °C in a H2850 Microwave Processor (EBSciences, East Granby, USA). After a washing step in phosphate-buffer saline containing 0.5% Tween (PBS-T), blocking was performed with 5% normal goat serum for 20 min. The antibody was diluted 1:200 in PBS-T and incubated overnight at 4 °C. The signal was detected with the Dako REAL Detection System (Dako Denmark A/S, Glostrup, Denmark) according to the manufacturer’s instruction. Finally, the slides were counterstained in Mayers Hämalaunlösung (Merck KGaA, Darmstadt, Germany) and mounted with Aquatex^®^ (Merck KGaA, Darmstadt, Germany) mounting medium. FFPE brain tissue slides of animal 31292 and 50000 served as negative controls.

### 4.10. In Situ Hybridization

Chromogenic in situ hybridization (ISH) was performed using the RNAscope system (Advanced Cell Diagnostics, Newark, NJ, USA). Probes were designed to target coding regions of the nucleocapsid protein (Advances Cell Diagnostics, Newark, NJ; Cat No. 533061) and latency-associated nuclear antigen (Advances Cell Diagnostics, Newark, NJ; Cat No. 571931) for PIV5 and BoHV6, respectively. A BoAstV CH13 RNAscope probe was already available (Advances Cell Diagnostics, Newark, NJ; Cat No. 406921). All available FFPE tissue slides of animals 26731, 27020, and 26875 were stained with the PIV5 probe. Tissue slides of animal 26875 and 27020 were additionally stained with the probe for BoAstV CH13 and tissue slides of animals 26731 and 27020 with the probe for BoHV6. Staining was performed with the RNAscope 2.5 Detection Kit-Brown (Advanced Cell Diagnostics, Newark, NJ) according to the manufacturer’s instructions. The slides were counterstained in Mayers Hämalaunlösung (Merck KGaA, Darmstadt, Germany) and mounted with Aquatex^®^ (Merck KGaA, Darmstadt, Germany) mounting media. FFPE brain tissue slides of animals 31292 (PIV5 staining), 44282 (BoAstV CH13 staining), 25018 (BoHV6 staining), and 50000 (PIV5 staining) served as negative controls.

### 4.11. Combination of In Situ Hybridization and Immunofluorescence

Fluorescent ISH using the RNAscope system (Advanced Cell Diagnostics, Newark, NJ) and the PIV5 probe (Advances Cell Diagnostics, Newark, NJ; Cat No. 533061) was performed on all FFPE slides of animal 26731, which were positive in chromogenic ISH. The RNAscope 2.5 Detection Kit-Red (Advanced Cell Diagnostics, Newark, NJ) was used according to the manufacturer’s instructions to the step of signal detection. To maintain tissue quality, the protocol provided by the manufacturer was adjusted by reducing the incubation time of protease pretreatment to 20 min. Then, immunofluorescence (IF) was performed under humid conditions with different cell markers. Therefore, the slides were washed in PBS-T and incubated with 10% normal goat serum for 20 min at room temperature. Afterwards, in every experiment, one primary antibody was added. Antibodies used were monoclonal mouse anti human neurofilaments clone 2F11 (Dako Denmark A/S, Glostrup, Denmark), polyclonal rabbit anti glial fibrillary acidic protein (GFAP) (Dako Denmark A/S, Glostrup, Denmark), polyclonal rabbit anti ionized calcium-binding adapter molecule 1 (Iba-1) (WAKO, Chuo-ku, Osaka, Japan), and monoclonal mouse anti 2′,3′-cyclic-nucleotide 3′-phosphodiesterase clone 5-11B (CNPase) (Thermo Fisher Scientific Inc., Waltham, USA) antibodies. Neurofilaments were incubated for 1 h at 37 °C diluted 1:100 in PBS-T, GFAP was incubated for 1 h at 37 °C diluted 1:1000 in PBS-T, Iba-1 was incubated at 37 °C for 1 h diluted 1:500 in PBS-T, and CNPase was incubated at 4 °C overnight diluted at 1:800 in PBS-T. Then, another washing step with PBS-T was performed before the secondary Alexa Fluor 488 goat anti-rabbit or goat anti-mouse antibody (Abcam plc, Cambridge, UK) diluted to 1:1000 in PBS-T together with DAPI BioChemica (AppliChem GmbH, Darmstadt, Germany) diluted to 1:10,000 in PBS-T were incubated for 1 h at room temperature. After washing in PBS-T and distilled water, the slides were mounted with Glycergel^®^, Aqueous Mounting Medium (Dako Denmark A/S, Glostrup, Denmark). Correlation analyses were performed on an Olympus Fluoview FV3000 Confocal Laser Scanning Microscope (Olympus Europa, Hamburg, Germany). FFPE slides of animal 31292 served as negative controls.

Semi-quantitative analysis of the infected cell types was performed on the midbrain of animal 26731. In three infected areas of FFPE brain tissue slides, which were double stained with fluorescent ISH for PIV5 and IF for different cell markers as described above, the number of each cell type was counted on an Olympus Fluoview FV3000 Confocal Laser Scanning Microscope (Olympus Europa, Hamburg, Germany) in a 60× magnification and a 1× and 2.46× zoom. Then, PIV5 infection of neurons (double staining with anti-neurofilaments, clone 2F11), astrocytes (double staining with anti-GFAP), microglia (double staining with anti-Iba-1), or oligodendrocytes (double staining with anti-CNPase, clone 5-11B) were counted in the same areas and the same settings. For each cell type, the percentage of PIV5 infection was calculated from each area and each setting and the mean value was determined. The proportion of infected cells for each cell type was scored in three categories: 1–30%, 31–60%, and 61–100%.

## Figures and Tables

**Figure 1 ijms-21-00498-f001:**
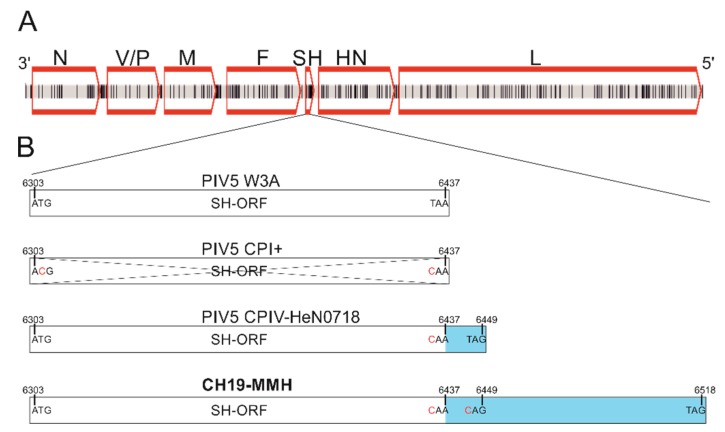
Schematic presentation of the PIV5 CH19-MMH genome and comparison to other PIV5 strains. (**A**) Nucleotide differences to the closest related PIV5 strain H221 are marked as black vertical lines. Open reading frames (ORFs) are indicated by red boxes. N: nucleocapsid protein, V: V-protein, P: phosphoprotein, M: matrix protein, F: fusion protein, SH: small hydrophobic protein, HN: hemagglutinin-neuraminidase protein, L: large protein. (**B**) The open reading frame of the SH protein (SH-ORF) varies in published full-length PIV5 genomes. In most strains, the SH-ORF is 132 bp long (exemplarily shown with strain W3A). In some strains, the start as well as the stop codon is mutated (exemplarily shown with strain CPI+) so that the gene is not probably transcribed. In two strains, a mutation in the stop codon is described, leading to a 12 bp longer ORF (exemplarily shown with strain CPIV-HeN0718). In the sequenced PIV5 CH19-MMH genome of this study, two stop codons are mutated, suggesting an 81 bp longer ORF. Numbers stand for the nucleotide position in the PIV5 genome, letters stand for bases, and red letters indicate mutated bases compared to the PIV5 reference strain. ORFs are depicted by boxes, the crossed-out box indicates a missing ORF, and the blue color indicates the ORF elongations.

**Figure 2 ijms-21-00498-f002:**
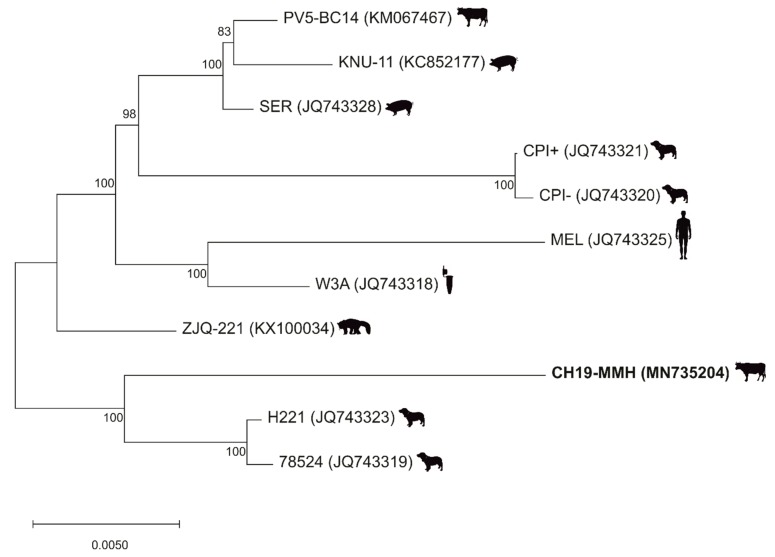
Phylogenetic analyses show an outstanding position of the PIV5 CH19-MMH strain. A PIV5 maximum likelihood tree based on selected full-length nucleotide sequences. PIV5 strain names are followed by accession numbers in brackets. The virus strain sequenced in this study is written in bold. The sources the strains were firstly described in are depicted by silhouettes behind the name and include cows, pigs, dogs, human, cell culture, and lesser panda.

**Figure 3 ijms-21-00498-f003:**
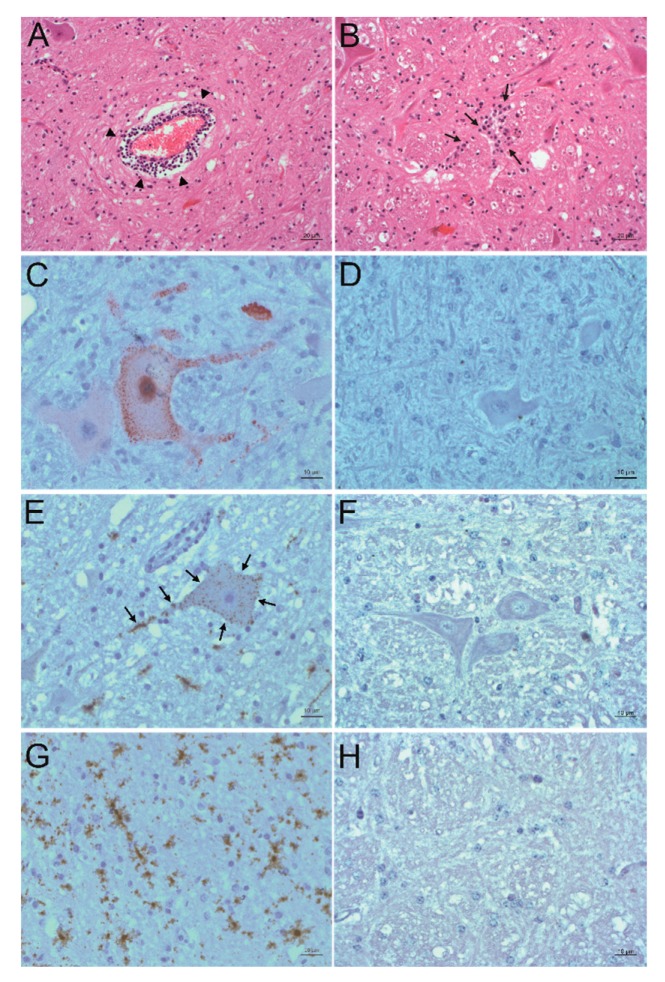
Detection of PIV5 antigen and RNA and their association to sites of inflammation. A, B: Histopathological analysis. Hematoxylin and eosin (H&E) staining of the formatio reticularis of animal 26731. Signs of non-suppurative encephalitis are present as perivascular cuffs (**A**, arrowheads) and glial nodes (**B**, arrows). C, D: Detection of viral antigen by immunohistochemistry (IHC) using the antibody PIV5 V5 TAG clone 2F11F7. In the formatio reticularis of animal 26731 (**C**), PIV5 antigen (red) can be seen in neurons, whereas in negative control animal 31292 (**D**), no PIV5 antigen can be detected. E, F: Detection of viral RNA by in situ hybridization (ISH) using the PIV5 RNAscope probe. In the formatio reticularis of animal 26731 (**E**), PIV5 RNA (arrows, brown) can be seen in neurons, whereas in negative control animal 31292 (**F**), no PIV5 RNA can be detected. (**G**,**H**) Detection of viral RNA with ISH using the PIV5 RNAscope probe. In the rhinal cortex of animal 26731 (**G**), PIV5 RNA (brown) shows no clear association to a specific cell type. In negative control animal 31292 (**H**), no PIV5 RNA can be detected. The scale bars at the bottom of the microphotographs correspond to 20 µm (**A**,**B**,**G**), and 10 µm (**C**–**F**,**H**), respectively.

**Figure 4 ijms-21-00498-f004:**
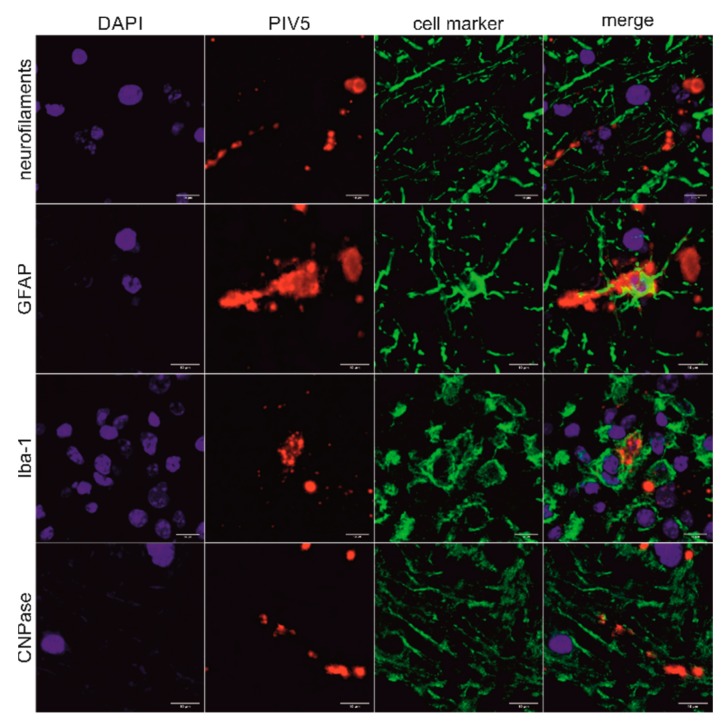
Correlation of PIV5 RNA with different brain cell types in animal 26731. Combination of fluorescent in situ hybridization (ISH) using the PIV5 RNAscope probe with immunofluorescence (IF) using different cell markers of the midbrain of animal 26731. The cell markers used were neurofilaments clone 2F11, glial fibrillary acidic protein (GFAP), ionized calcium-binding adapter molecule 1 (Iba-1), and 2′,3′-cyclic-nucleotide 3′-phosphodiesterase clone 11-5B (CNPase). Nuclei were stained in blue, PIV5 RNA was stained in red, and different cell markers were stained in green. A clear association of PIV5 RNA with axons, astrocytes, microglia, and myelin sheaths can be seen. Scale bars: 10 µm.

**Figure 5 ijms-21-00498-f005:**
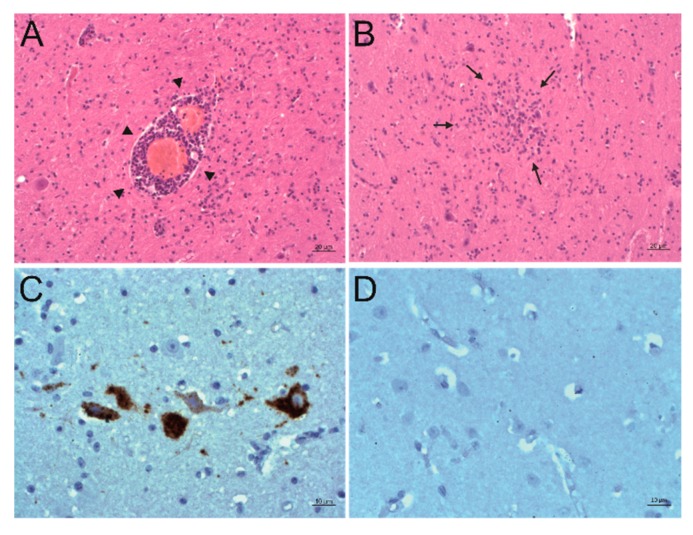
Correlation of BoAstV CH13 RNA to sites of inflammation. (**A**,**B**) Histopathological analysis. Hematoxylin and eosin (H&E) staining of the midbrain of animal 26875. Signs of non-suppurative encephalitis are present as perivascular cuffs (**A**, arrowheads) and glial nodes (**B**, arrows). (**C**,**D**) Detection of viral RNA with in situ hybridization (ISH) using the BoAstV RNAscope probe. In the midbrain of animal 26875 (**C**), BoAstV CH13 RNA (brown) can be seen in neurons, whereas in negative control animal 44285 (**D**), no BoAstV CH13 RNA can be detected. Scale bars: 20 µm (**A**,**B**) and 10 µm (**C**,**D**).

**Table 1 ijms-21-00498-t001:** History and initial testing results of the three PIV5-positive animals from previous studies.

History	26731	Animal 26875	27020
**Date of receipt**	27 February 1998	28 April 1998	10 July 1998
**Age (years) at time of death**	5	2	3
**Reported neuro-logical** **symptoms**	stall walking	peracute: tremor of head and neck, reduced vision, hypersensitivity in hind limbs	neurological symptoms for 3 weeks
**Histological examination at time of admission**	- meningo-encephalitis - mononuclear, perivascular infiltrations - gliosis - neuronal necrosis - location: basal ganglia, diencephalon and mesencephalon	- meningo-encephalitis - mononuclear infiltrations (and mononuclear meningitis) - gliosis - neuronal necrosis - location: cerebellum and brainstem	- meningo-encephalitis - mononuclear infiltrations (and mononuclear meningitis) - gliosis - location: cortex, brainstem (especially mesencephalon) and cerebellum
**Studies PIV5 was detected**	Wüthrich et al., *Virology* (2016) ^1^	Bouzalas et al., *Infect Genet Evol* (2016) ^2^	Wüthrich et al., *Virology* (2016) ^1^
**Next Generation Sequencing results for PIV5**	+ ^1^ low read depth and low coverage	+ ^2^ high read depth and nearly complete coverage	- ^1^
**RT-PCR results for PIV5**	+ ^1^	not performed	+ ^1^
**Other viruses found in the brain (method)**	BoHV6 (qPCR ^1^), BoPyV2 (PCR ^1^)	BoAstV-CH13 (RT-PCR ^3^, NGS^2^, ISH ^3^, IHC ^4^, RT-qPCR ^5^)	BoHV6 (qPCR ^1^), BoPyV2 (PCR^1^), BoAstV CH13 (ISH ^6^, RT-qPCR ^5^)

^1^ Wüthrich et al., Virology (2016) [27]; ^2^ Bouzalas et al., Infect Genet Evol (2016) [28]; ^3^ Bouzalas et al., J Clin Microbiol (2014) [24], ^4^ Boujon et al., J Virol Methods (2017) [45], ^5^ Lüthi et al., Sci Rep (2018) [44], ^6^ Selimovic-Hamza et al., Viruses (2017) [26]; “+”: positive result; “-”: negative result; “PIV5”: parainfluenza virus 5; “BoHV6”: bovine herpesvirus 6; “BoPyV2”: bovine polyomavirus 2; “BoAstV CH13”: bovine astrovirus CH13; “NGS”: Next Generation Sequencing; “ISH”: in situ hybridization; “IHC”: immunohistochemistry.

**Table 2 ijms-21-00498-t002:** Copy numbers of PIV5, BoAstV CH13, and BoHV6 in animals 26731, 26875, and 27020.

Virus	26731	Animal 26875	27020
**PIV5 ^A^**	4.09 × 10^4^	1.87 × 10^2^	1.11 × 10^1^
**BoAstV CH13 ^A^**	-	5.74 × 10^5^	5.14 × 10^2^
**BoHV6 ^B^**	2.79 × 10^2^	-	1.58 × 10^2^

^A^ per 1 μL RNA; ^B^ per 3.75 μL DNA

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
