# Peer review of "Parainfluenza Virus 5 Infection in Neurological Disease and Encephalitis of Cattle"

_ijms, 2020, doi:10.3390/ijms21020498_

Round 1

Reviewer 1 Report

This retrospective study is part of a larger effort to resolve past cases of bovine encephalitis of unknown etiology.  Previous work from this group focused on using unbiased sequence analysis to screen clinical samples and identified bovine astrovirus CH13, which was later found in several cattle with non-suppurative encephalitis.  Here, evidence is provided that parainfluenza virus 5 (PIV5) can also contribute to encephalitis in cattle.  Complete (or near-complete) genome sequencing identified a novel PIV5 strain (PIV5 CH19-MMH) associated with all three bovine samples testing positive for the virus.  Interestingly, the SH ORF of PIV5 CH19-MMH is substantially longer than the SH ORFs of other PIV5 strains, although the impact of the longer ORF on virus replication and pathogenesis remains to be defined.  In one of the three brain samples, PIV5 antigen and RNA was detected in situ with correlation to sites of inflammation, thereby defining PIV5 infection as a probable cause of the neurological disease.

This is a well-written manuscript with clear data that substantially advances understanding of the probable causes of bovine encephalitis.  I have only minor concerns.

p. 6, lines 16-18 “According to a semi-quantitative evaluation, infection of neurons takes place frequently, whereas infection of astrocytes is found only seldom. Oligodendrocytes and microglia were also found infected with PIV5 but only occasionally”.  Please provide more detail on how this conclusion was arrived at.  It is not clear from Fig. 4.  How was the semi-quantitative evaluation performed?

   2. p. 2, line 18 “non-suppruative”

p. 6, line 19-20  “…it was not possible to detect neither PIV5 antigen, nor RNA”.  Please re-write for grammar and clarity. p. 9, line 8 “In phylogenetic analyses, however, the closest related strain is the respiratory dog PIV5 H221 strain and are not other neurotropic or bovine PIV5s underlining the hypothesis of a new virus origin.”.  Please re-write for grammar and clarity.

Author Response

Point: "p. 6, lines 16-18 “According to a semi-quantitative evaluation, infection of neurons takes place frequently, whereas infection of astrocytes is found only seldom. Oligodendrocytes and microglia were also found infected with PIV5 but only occasionally”.  Please provide more detail on how this conclusion was arrived at.  It is not clear from Fig. 4.  How was the semi-quantitative evaluation performed?"

Response: Thanks for this comment. We agree that this was not clearly explained. We have added a paragraph in the materials and methods section and modified the results section accordingly.

Point: "2. p. 2, line 18 “non-suppurative”

Response: is corrected

Point: "p. 6, line 19-20  “…it was not possible to detect neither PIV5 antigen, nor RNA”.  Please re-write for grammar and clarity. "

Response: Thank you! We corrected this sentence "… it was not possible to detect either PIV5 antigen, or RNA"

Point: p. 9, line 8 “In phylogenetic analyses, however, the closest related strain is the respiratory dog PIV5 H221 strain and are not other neurotropic or bovine PIV5s underlining the hypothesis of a new virus origin.”.  Please re-write for grammar and clarity."

Response: We agree that this statement was not clear enough. We have rephrased this section.

Reviewer 2 Report

Hierweger et al. analyzed the possible role of parainfluenza virus 5 (PIV) infection in bovine encephalomyelitis using brain samples obtained for exclusion of BSE in a specific Division, University of Bern, in 1998. They have performed a detailed analysis for three cases, which were positive for PIV5 by PCR or NGS, using formalin-fixed paraffin-embedded brain tissues. The in situ analysis of brain tissues of one case (animal #26731) showed positive results for viral RNA and viral antigens in the pathological region with inflammation, while that of the other two cases (#26875 and #27020) was negative at least for the samples used in this study. In general, the study has been performed well, and the manuscript has been written under the careful consideration, avoiding strong assumptions. This reviewer has only several minor comments for authors’ consideration for improvement of the manuscript.

Minor comments

Certain variations in the nucleotide sequences and ORF length of viral proteins exist for the detected PIV5 strain. Thus, authors have thus stated that the strain genome has unique features (in abstract) or that the virus sequences differ from already published PIV5 sequences (line 19, page 8). On the other hand, the authors describe that there are only little sequence variation (line 19, page 5) or that the overall very little sequence variations (line 11, page 9). These descriptions may be inconsistent.

Only brain tissues were analyzed in this study. Thus, ‘a broad cell tropism’ of the PIV5 strain cannot be assessed in this study. (Abstract, line 20).

If available, data of serology and other organs are quite useful to understand the role of PIV5 infection in these animals.

Please check the consistency between the main text (lines 8–18, page 3) and Table 1. Table 1 indicates that PCR testing has not been conducted for the animal #26875, while the main text describes that the presence of PIV5 was confirmed with PCR in all cases. Similarly, Table 1 indicates that BoPyV2 was positive for #26731 and #27020 by PCR, while the main text describes that it was not possible to confirm the presence of BoPyV2 with PCR in any of the samples.

For what negative controls were they? Does it mean negative samples for PIV5? (line 15)

This reviewer cannot understand the authors’ message in the sentences in lines 16–19, page 3 (Animal brain tissue samples that were ~).

Please check the sentence (The ORF of the F protein~) in line 26–27. The ORF of the F protein my be 66nt longer, but not shorter, than PIV5 isolates from cell culture.

Was the PIV5 RNA detected outside the cells? (the figure legend for Figure 3 (G), line 10–11, page 7)

Please describe the more detail of the antibody (clone 2F11F7) specific for the PIV5 V protein in the main text of the result section, where the immunohistochemistory data are discussed.

Author Response

Point: "Certain variations in the nucleotide sequences and ORF length of viral proteins exist for the detected PIV5 strain. Thus, authors have thus stated that the strain genome has unique features (in abstract) or that the virus sequences differ from already published PIV5 sequences (line 19, page 8). On the other hand, the authors describe that there are only little sequence variation (line 19, page 5) or that the overall very little sequence variations (line 11, page 9). These descriptions may be inconsistent.

Response: Thanks for this comment. Although there is very little sequence variations between PIV 5 strains on the pure nt level, these little variations may have larger impact as the affect start and stop codons (as for the SH-ORF) and thus result in considerably modified viral proteins. In this regard we do not agree that our statements are inconsistent, however, for clarification we have changed this point in the discussion. 

 Point :"Only brain tissues were analyzed in this study. Thus, ‘a broad cell tropism’ of the PIV5 strain cannot be assessed in this study. (Abstract, line 20).

Response: Thanks, we agree with this point and change this sentence in the abstract to: " …a broad cell tropism of PIV5 in the brain."

Point :"If available, data of serology and other organs are quite useful to understand the role of PIV5 infection in these animals."

Response: Thank you! We fully agree with this point. However, serum samples and other organ samples of the animals under investigations were not available. A broader serological survey would be very useful to assess the overall prevalence of PIV5 infection in the cattle population. This is planned for future studies, but was beyond the scope of the present work.  A statement on this direction is part of the discussion.

Point:" Please check the consistency between the main text (lines 8–18, page 3) and Table 1. Table 1 indicates that PCR testing has not been conducted for the animal #26875, while the main text describes that the presence of PIV5 was confirmed with PCR in all cases. Similarly, Table 1 indicates that BoPyV2 was positive for #26731 and #27020 by PCR, while the main text describes that it was not possible to confirm the presence of BoPyV2 with PCR in any of the samples.

Thank you. There is indeed inconsistency between previous studies and the present study regarding BoPyV2 detection. It was not clear that Table 1 compiles the results of previous studies, which we now explain in the table title. Moreover, we have changed section 2.1 for clarity.  

Point:"For what negative controls were they? Does it mean negative samples for PIV5? (line 15)"

Response: We have added this information: "Brain tissue samples of cattle with non-suppurative encephalitis of another origin than PIV5, BoAstV CH13 or BoHV6 served as negative controls…"

Point: "This reviewer cannot understand the authors’ message in the sentences in lines 16–19, page 3 (Animal brain tissue samples that were ~)."

Response: Thanks. We have added on page 3, line 17: "In order to exclude carryover of PIV5 during brain tissue preparation between animals 26731, 26875 and 27020…"

Point: "Please check the sentence (The ORF of the F protein~) in line 26–27. The ORF of the F protein my be 66nt longer, but not shorter, than PIV5 isolates from cell culture."

Response: This information was incorrect and we have corrected it.

Point: Was the PIV5 RNA detected outside the cells? (the figure legend for Figure 3 (G), line 10–11, page 7)

Response: We have changed wording to "PIV 5 RNA shows no clear association to a specific cell type"

Point: Please describe the more detail of the antibody (clone 2F11F7) specific for the PIV5 V protein in the main text of the result section, where the immunohistochemistory data are discussed.

Reponse: done